# CALIBRATING SEQUENCE LIKELIHOOD IMPROVES CONDITIONAL LANGUAGE GENERATION

**Yao Zhao**
yaozhaoyz@google.com

**Misha Khalman**
khalman@google.com

**Rishabh Joshi**
rishabhjoshi@google.com

**Shashi Narayan**
shashinarayan@google.com

**Mohammad Saleh**
msaleh@google.com

**Peter J. Liu**
peterjliu@google.com

Google Research, Brain Team

## ABSTRACT

Conditional language models are predominantly trained with maximum likelihood estimation (MLE), giving probability mass to sparsely observed target sequences. While MLE trained models assign high probability to plausible sequences given the context, the model probabilities often do not accurately rank-order generated sequences by quality. This has been empirically observed in beam search decoding as output quality degrading with large beam sizes, and decoding strategies benefiting from heuristics such as length normalization and repetition-blocking. In this work, we introduce *sequence likelihood calibration* (SLiC) where the likelihood of model generated sequences are calibrated to better align with reference sequences in the model's latent space. With SLiC, decoding heuristics become unnecessary and decoding candidates' quality significantly improves regardless of the decoding method. Furthermore, SLiC shows no sign of diminishing returns with model scale, and presents alternative ways to improve quality with limited training and inference budgets. With SLiC, we exceed or match SOTA results on a wide range of generation tasks spanning abstractive summarization, question generation, abstractive question answering and data-to-text generation, even with modest-sized models.

## 1 INTRODUCTION

Conditional language generation aims to generate text based on input context, and includes many useful and hard tasks such as abstractive summarization (Mani, 2001; Nenkova and McKeown, 2011), generative question answering (Bajaj et al., 2016), question generation (Zhou et al., 2017) and data-to-text (Wiseman et al., 2017; Gardent et al., 2017). Pretraining large Transformer encoder-decoder models and fine-tuning them on downstream tasks is the common paradigm to address these tasks (Raffel et al., 2020; Lewis et al., 2019; Tay et al., 2022; Zhang et al., 2019a).

Conditional language generation tasks are modeled by learning the probability of a target sequence $\mathbf{y}$ given a context sequence $\mathbf{x}$. Since directly modeling sequence probability $P(\mathbf{y}|\mathbf{x})$ over all possible generated text sequences is intractable, the canonical solution is to auto-regressively factor the probability and share the parameters at all token prediction steps as $P_\theta(\mathbf{y}|\mathbf{x}) = \prod_{t=0}^{l} P_\theta(y^t|y^0...y^{t-1}, \mathbf{x})$, where $l$ is the sequence length. These models are often trained with maximum likelihood estimation (MLE) over observed target sequences. The learning objective thus becomes $L = \sum_i^N -\log P_\theta(\mathbf{y}_i|\mathbf{x}_i) = \sum_i^N \sum_{t=0}^{l} -\log P_\theta(y_i^t|y_i^0...y_i^{t-1}, \mathbf{x}_i)$, where $N$ is the number of training instances. It is also referred to as next token prediction loss.

In the ideal setting of MLE training, a large number of target sequences are observed for each context, and the relative frequencies of output sequences can calibrate the assigned model probabilities. However, in practice most language generation training datasets have only a single target sequence given the context. While the subsequent MLE trained models learn to assign relatively high probability to plausible sequences, they lack the direct supervision to compare such sequences, and solely

Figure 1: Calibrating sequence likelihood improves language generation across model scales. Scores are averaged ROUGE across 4 datasets ($R_m$ in subsection 3.2)

rely on models' generalization capability. We refer to this phenomenon as models' sequence likelihood not being *calibrated*. Prior works (Liu and Liu, 2021; Liu et al., 2022) has shown that the correlation between sequence probability and its quality for MLE trained models can be low. Liu et al. (2022) attributed this similarly as the deterministic (one-point) target distribution problem. Exposure bias (Ranzato et al., 2016) further aggravates the problem, as sequence likelihood estimation is noisier when models' decoded sequences shift from exposed training data distribution.

Many effective heuristics have been proposed during training and decoding to combat the problem of uncalibrated sequence likelihood. Label smoothing (Szegedy et al., 2016) prevents the network from becoming over-confident towards the observed target. This is particularly necessary in language generation, since the gold target represents just one of many possibilities. It has been observed that increasing number of decoding candidates past a certain point leads to worse quality for beam search decoding (Yang et al., 2018; Koehn and Knowles, 2017) and sampling (Adiwardana et al., 2020). An optimal number of decoding candidates is often determined empirically by decoding models on the validation set and measuring their performance. Using length normalization is also essential for beam search decoding (Wu et al., 2016) and sampling (Adiwardana et al., 2020) as models tend to underestimate sequence likelihood of longer sentences. Repetition is another common failure mode when models overestimate the probability of repeated sequences (Holtzman et al., 2019). Trigram blocking (Paulus et al., 2018) and nucleus sampling (Holtzman et al., 2020) have been used to interrupt repeating sequences. These techniques are pervasive and often the default in modern Transformer libraries (Wolf et al., 2020; Lewis et al., 2019; Raffel et al., 2020; Zhang et al., 2019a).

Since the lack of observed target sequences in MLE training is the root problem, solutions involving learning with multiple sequence candidates have been proposed to directly address it. They can be loosely put in three categories: (1) reinforcement learning with sequence-level rewards (Paulus et al., 2018; Ziegler et al., 2019; Stiennon et al., 2020); (2) two-stage systems that generate and rerank candidates (Liu and Liu, 2021; Ravaut et al., 2022b; Liu et al., 2022); and (3) multi-task learning with sequence-level losses (Edunov et al., 2018; Liu et al., 2022). Refer to Related Works (section 4) for a more comprehensive discussion.

In this paper, we propose to first decode candidates from a fine-tuned model on its own training dataset, and then continue training the model with a new objective. The new objective aims to align candidates' sequence likelihoods according to their similarities to the target sequence in the model's latent space. We refer to this process as **sequence likelihood calibration** (SLiC). Our approach is related to multi-task learning with sequence-level losses in Liu et al. (2022). However, we propose a simple yet effective recipe that eliminates decoding heuristics and doesn't risk directly optimizing the same metrics that are used to report text generation quality. Unlike reinforcement learning, it is a one-time offline process that avoids costly online decoding processes. Also, when compared to two-stage reranking systems, it doesn't require a separate reranking model that incurs additional complexity and compute. As depicted in Figure 1, our calibration stage naturally extends the current paradigm of pretraining and fine-tuning, and we show that calibrated models have strong improvements over fine-tuned-only models across model sizes.

Our main contributions include:

- Proposed a sequence likelihood calibration (SLiC) stage that consistently improves model quality, exceeding or matching state-of-the-art results on abstractive summarization, generative question answering, question generation and data-to-text generation tasks.

- Proposed a novel calibration similarity metric between model decodes and targets measured in the model's latent space rather than resorting to external metrics or human feedback.

- Demonstrated that SLiC eliminates the need for popular decoding heuristics, such as beam size optimization, length normalization and repetition prevention for the calibrated models.

- Demonstrated that SLiC has persistent significant benefits on model performance even as the number of model parameters scales up. Under the same inference budget, smaller calibrated models might outperform larger counterparts by decoding more candidates.

## 2 CALIBRATING SEQUENCE LIKELIHOOD

We extend the common paradigm of pretraining and fine-tuning by introducing a third calibration stage, SLiC. As shown in Algorithm 1, we first decode $m$ candidates $\{\hat{\mathbf{y}}\}_m$ from a fine-tuned model $P_{\theta_{ft}}(\mathbf{y}|\mathbf{x})$ on fine-tuning dataset $\{\mathbf{x}, \bar{\mathbf{y}}\}_n$ and then calibrate the fine-tuned model by continuing training on our proposed loss: $\mathcal{L}(\theta) = \sum_b L^{\mathrm{cal}}(\theta, s; \mathbf{x}, \bar{\mathbf{y}}, \{\hat{\mathbf{y}}\}_m) + \lambda L^{\mathrm{reg}}(\theta, \theta_{ft}; \mathbf{x}, \bar{\mathbf{y}})$ , where $\theta$ and $\theta_{ft}$ are the current and fine-tuned-only model weights, $L^{\mathrm{cal}}$ and $L^{\mathrm{reg}}$ are the calibration and regularization losses. $s = s(\hat{\mathbf{y}}, \bar{\mathbf{y}}; \mathbf{x})$ measures the similarity between the candidate $\hat{\mathbf{y}}$ and the target $\bar{\mathbf{y}}$ conditioned on the context $\mathbf{x}$. We discuss choices of $s$, $L^{\mathrm{cal}}$, $L^{\mathrm{reg}}$ and decode strategies $\hat{\mathbf{y}} \sim P_\theta(\mathbf{y}|\mathbf{x})$ in the following sections.

---

**Algorithm 1** Calibrating Sequence Likelihood

$\quad$ **for** $\mathbf{x}, \bar{\mathbf{y}} \in \{\mathbf{x}, \bar{\mathbf{y}}\}_n$ **do** $\qquad\qquad\qquad\qquad$ ▷ sample $m$ candidates from the fine-tuned model
$\qquad \{\hat{\mathbf{y}} \sim P_{\theta_{ft}}(\mathbf{y}|\mathbf{x})\}_m$

$\quad \theta \leftarrow \theta_{ft} \qquad\qquad\qquad\qquad\qquad\qquad\qquad$ ▷ initialized from the fine-tuned model
$\quad$ **for** $\{\mathbf{x}, \bar{\mathbf{y}}, \{\hat{\mathbf{y}}\}_m\}_b \sim \{\mathbf{x}, \bar{\mathbf{y}}, \{\hat{\mathbf{y}}\}_m\}_n$ **do** $\quad$ ▷ train with calibration and regularization loss
$\qquad \theta \leftarrow \theta - lr\nabla_\theta \mathcal{L}(\theta)$

---

### 2.1 SIMILARITY FUNCTION: $s$

For a given output sequence $\mathbf{y}$, we take the decoder output hidden states $\mathbf{e}^{L \times D} = \mathrm{emb}(\mathbf{y}, \mathbf{x})$ as its representations, where $L$ is the number of tokens and $D$ is the hidden states dimension. Between a candidate $\hat{\mathbf{y}}$'s representations $\hat{\mathbf{e}}$ and the target $\bar{\mathbf{y}}$'s representations $\bar{\mathbf{e}}$, we calculate their cosine similarities on spans of $n$ tokens and aggregate them across the sequences with a F-measured based function $F_n$. Notation of $F_n, P_n, R_n$ are same as in BERTScore (Zhang et al., 2019b).

$$s_\theta(\hat{\mathbf{y}}, \bar{\mathbf{y}}; \mathbf{x}) = \sum_n F_n(\hat{\mathbf{e}}, \bar{\mathbf{e}}) = \sum_n F_n(\mathrm{emb}(\hat{\mathbf{y}}, \mathbf{x}), \mathrm{emb}(\bar{\mathbf{y}}, \mathbf{x})) \qquad F_n = 2\frac{P_n \times R_n}{P_n + R_n}$$

$$P_n(\hat{\mathbf{e}}, \bar{\mathbf{e}}) = \frac{1}{|\hat{\mathbf{e}}|} \sum_{\hat{\mathbf{e}}_{i:i+n}} \max_{\bar{\mathbf{e}}_{j:j+n}} \hat{\mathbf{e}}_{i:i+n}^T \bar{\mathbf{e}}_{j:j+n} \qquad R_n(\hat{\mathbf{e}}, \bar{\mathbf{e}}) = \frac{1}{|\bar{\mathbf{e}}|} \sum_{\bar{\mathbf{e}}_{j:j+n}} \max_{\hat{\mathbf{e}}_{i:i+n}} \hat{\mathbf{e}}_{i:i+n}^T \bar{\mathbf{e}}_{j:j+n}$$

Compared to BERTScore, we use our models' decoder output representations instead of BERT encoder representations and also consider matching on spans of $n = 1, 2, 4, 8$ tokens rather than 1.

Compared to using external metrics, such as ROUGE, BERTScore, this scoring function has a few advantages: (1) it adds very little compute cost, does not require extra model or out-of-graph computation; (2) it differs from the metrics that we evaluate the generation systems with and mitigates the risk of directly optimizing towards those imperfect metrics (Paulus et al., 2018; Stiennon et al., 2020); (3) it is conditioned on the context $s(\hat{\mathbf{y}}, \bar{\mathbf{y}}; \mathbf{x})$, as opposed to metrics in the form of $s(\hat{\mathbf{y}}, \bar{\mathbf{y}})$.

### 2.2 CALIBRATION LOSS: $L^{\mathrm{cal}}$

The calibration loss $L^{\mathrm{cal}}(\theta, s; \mathbf{x}, \bar{\mathbf{y}}, \{\hat{\mathbf{y}}\}_m)$ aims to align models' decoded candidates' sequence likelihood $P_\theta(\hat{\mathbf{y}}|\mathbf{x})$ according to their similarity with the target sequence $s(\hat{\mathbf{y}}, \bar{\mathbf{y}}; \mathbf{x})$. Given the context $\mathbf{x}$, target $\bar{\mathbf{y}}$ and a set of candidates $\{\hat{\mathbf{y}}\}_m$, we consider the following 4 loss types to answer two questions: (1) does absolute difference in similarities matter? (2) is there benefit of list-wise over pair-wise comparisons? **Rank** loss optimizes the ranking order of positive and negative candidates pairs $\hat{\mathbf{y}}_+, \hat{\mathbf{y}}_-$ uniformly sampled from $\{\hat{\mathbf{y}}\}_m$ where $s(\hat{\mathbf{y}}_+, \bar{\mathbf{y}}; \mathbf{x}) > s(\hat{\mathbf{y}}_-, \bar{\mathbf{y}}; \mathbf{x})$. **Margin** loss maximizes the sequence probability gap of positive and negative candidates pairs. **List-wise rank** loss

optimizes the ranking orders of a list of candidates, where $i, j$ are positions of $\hat{\mathbf{y}}_i, \hat{\mathbf{y}}_j$ in the set $\{\hat{\mathbf{y}}\}_m$ sorted by $s(\hat{\mathbf{y}}, \bar{\mathbf{y}}; \mathbf{x})$. List-wise rank loss is the contrastive loss used in BRIO (Liu et al., 2022). **Expected reward** loss (or expected minimum risk) maximizes the expected similarity of a list of candidates (Edunov et al., 2018). Pair-wise losses (Rank, Margin) has smaller training memory footprint than list-wise rank and expected reward.

$$
\begin{aligned}
L_{\text{rank}}^{\text{cal}} &= \max(0, \beta - \log P_\theta(\hat{\mathbf{y}}_+|\mathbf{x}) + \log P_\theta(\hat{\mathbf{y}}_-|\mathbf{x})) \\
L_{\text{margin}}^{\text{cal}} &= \max(0, \beta(s(\hat{\mathbf{y}}_+, \bar{\mathbf{y}}; \mathbf{x}) - s(\hat{\mathbf{y}}_-, \bar{\mathbf{y}}; \mathbf{x})) - \log P_\theta(\hat{\mathbf{y}}_+|\mathbf{x}) + \log P_\theta(\hat{\mathbf{y}}_-|\mathbf{x})) \\
L_{\text{list rank}}^{\text{cal}} &= \Sigma_{i<j} \max(0, \beta|i-j| - \log P_\theta(\hat{\mathbf{y}}_i|\mathbf{x}) + \log P_\theta(\hat{\mathbf{y}}_j|\mathbf{x})) \\
L_{\text{reward}}^{\text{cal}} &= \Sigma_i \left[ -s(\hat{\mathbf{y}}_i, \bar{\mathbf{y}}; \mathbf{x}) * \frac{P_\theta(\hat{\mathbf{y}}_i|\mathbf{x})}{\sum_i P_\theta(\hat{\mathbf{y}}_i|\mathbf{x})} \right]
\end{aligned}
\tag{1}
$$

$\beta$ values for all losses are chosen empirically for each loss type in subsection 3.3.

## 2.3 REGULARIZATION LOSS: $L^{\text{reg}}$

We consider two alternate types of regularization loss $L^{\text{reg}}$ to prevent models from deviating significantly from their fine-tuned MLE objective: **Cross entropy** is the standard fine-tuning MLE objective used in (Liu et al., 2022). **KL divergence** directly minimizes the probability distribution distance between the calibrated model and the fine-tuned model at each token on observed target sequence. The main difference is cross entropy loss regularizes the model toward the gold reference while KL divergence regularize the model toward fine-tuned-only model. The regularization losses are both on token level.

$$
L_{\text{ce}}^{\text{reg}} = \sum_t -\log P_\theta(\bar{y}_t|\bar{\mathbf{y}}_{t-1}, \mathbf{x}) \qquad L_{\text{kl}}^{\text{reg}} = \sum_t P_\theta(\bar{y}_t|\bar{\mathbf{y}}_{t-1}, \mathbf{x}) \log \frac{P_\theta(\bar{y}_t|\bar{\mathbf{y}}_{t-1}, \mathbf{x})}{P_{\theta_{ft}}(\bar{y}_t|\bar{\mathbf{y}}_{t-1}, \mathbf{x})}
\tag{2}
$$

## 2.4 CANDIDATES DECODING METHODS

We consider the following decoding methods for SLiC:

**Beam Search** is the standard best-first algorithm to solve the intractable maximum likelihood optimization for sequence-to-sequence models (Tillmann and Ney, 2003; Li et al., 2016; Wiseman et al., 2017; Chen et al., 2018).

**Diverse Beam Search** (DBS; Vijayakumar et al., 2016) generates a list of diverse outputs by dividing the beam search budget into groups and enforcing dissimilarity between groups of beams. It strikes balance between quality and diversity and is often the best strategy for two-stage reranking systems (Liu and Liu, 2021; Ravaut et al., 2022b; Liu et al., 2022).

**Nucleus Sampling** (Holtzman et al., 2020) only samples high-probable tokens within cumulative probability $p$ at each step of the decoding. It produces diverse candidates while preventing sampling very low quality ones.

## 3 EXPERIMENTS

### 3.1 TASKS AND DATASETS

For abstractive summarization tasks, we choose **CNN/DailyMail** (Hermann et al., 2015; See et al., 2017), **XSUM** (Narayan et al., 2018), **RedditTIFU-long** (Kim et al., 2019) and **SAMSum** (Gliwa et al., 2019) due to their diversity in domain, style, abstractiveness, and summary lengths. For question answering related tasks, we choose generative question answering given context **MSMARCO NLG** (Bajaj et al., 2016) and its reverse problem of question generation **SQuAD QG** (Zhou et al., 2017; Du et al., 2017). For data-to-text tasks, we choose text generation given structured data **WebNLG-en** (Gardent et al., 2017) and common concepts reasoning **CommonGen** (Lin et al., 2020). More details of datasets can be found at Appendix A along with their statistics.

## 3.2 MODEL TRAINING AND EVALUATION DETAILS

We follow the PEGASUS pretraining (Zhang et al., 2019a) and extend transformer model sizes to PEGASUS$_{\text{SMALL}}$ (50M), PEGASUS$_{\text{BASE}}$ (200M), PEGASUS$_{\text{LARGE}}$ (500M) and PEGASUS$_{\text{2B}}$ (2B). Details are reported in Appendix B. Different from the original paper, we use a sentencepiece 96k vocabulary with byte-fallback (Kudo, 2018) and pretraining batch size of $4096$ across all models. See Appendix B for model dimensions.

In all experiments, we use learning rate $lr = 10^{-4}$, and batch sizes of 512 to finetune and 64 to calibrate models. We use beam search to generate calibration candidates and evaluate the calibrated models, unless specified otherwise. All fine-tuned-only models utilizes heuristics such as beam size optimization and sweeping beam $\alpha$ for length normalization, unless specified otherwise.

In our ablation studies (subsection 3.3), benefits analysis (subsection 3.4), and scaling experiments (subsection 3.5), we use models pretrained to $500,000$ steps and conduct experiments on 4 datasets (CNN/DailyMail, XSUM, RedditTIFU-long and SAMSum). For ablation studies and benefits analysis, we use PEGASUS$_{\text{LARGE}}$. We report ROUGE 1/2/L (Lin, 2004) for each dataset on **validation** splits and their overall score $R_m$ defined as geometric mean of ROUGE 1/2/L averaged across datasets, $R_m = \frac{1}{4} \sum_d \sqrt[3]{R_1 R_2 R_L}$.

For the final results (subsection 3.6), we pretrain PEGASUS$_{\text{2B}}$ model to $2.5M$ steps, fine-tune it on all 8 datasets, calibrate them using the same recipe and report numbers on the **test** split (unless specified otherwise). We use corresponding standard evaluation scripts for each dataset (details in Appendix D).

## 3.3 ABLATION STUDIES OF CALIBRATION

Ablation experiment results discussed below can be found in Table 1.

Table 1: Ablation of the sequence likelihood calibration method. Shared hyper-parameters are held as constant within each comparison group but vary between groups (Appendix E). $\Delta$ is the relative improvements of overall score $R_m$ compared with the fine-tuned model. Choices in our recommend recipe are in bold.

| Ablation | CNN/DailyMail R1 / R2 / RL | XSUM R1 / R2 / RL | RedditTIFU-long R1 / R2 / RL | SAMSum R1 / R2 / RL | $\Delta$ avg |
|---|---|---|---|---|---|
| fine-tuned | 44.74/21.83/41.92 | 47.23/24.31/39.12 | 26.84/9.08/21.92 | 53.67/29.35/44.75 | 0.00% |
| *similarity function* | | | | | |
| ROUGE | 46.47/22.49/43.63 | 47.86/24.55/39.58 | 29.92/9.83/23.93 | 54.82/30.15/45.30 | 3.26% |
| **decoder repr** | 46.55/22.50/43.69 | 47.88/24.62/39.62 | 29.86/9.84/23.91 | 54.72/29.96/45.10 | 3.20% |
| token emb | 46.51/22.48/43.67 | 47.04/23.63/38.39 | 29.78/9.69/23.46 | 53.71/29.38/44.75 | 1.64% |
| *calibration loss* | | | | | |
| **rank** | 46.73/22.70/43.85 | 48.11/24.80/40.06 | 30.34/9.80/24.32 | 55.19/30.46/46.32 | 4.27% |
| margin | 46.11/22.46/43.30 | 47.62/24.81/39.89 | 30.84/9.97/24.37 | 54.58/30.10/45.92 | 3.63% |
| list rank | 46.62/22.88/43.76 | 47.93/24.57/39.67 | 30.87/9.65/24.46 | 54.56/29.81/45.17 | 3.49% |
| reward | 46.49/22.55/43.63 | 47.77/24.48/39.49 | 30.99/9.95/24.39 | 54.42/29.98/45.56 | 3.47% |
| *regularization loss* | | | | | |
| none | 46.54/22.44/43.68 | 47.51/24.70/39.82 | 30.73/9.68/24.05 | 55.07/30.07/45.60 | 3.48% |
| cross entropy | 46.73/22.70/43.85 | 48.11/24.80/40.06 | 29.96/9.72/23.82 | 55.19/30.46/46.32 | 4.06% |
| **KL divergence** | 46.80/22.83/43.98 | 47.96/24.92/40.09 | 30.73/9.68/24.05 | 54.87/30.20/45.95 | 4.09% |
| *candidates decoding method* | | | | | |
| **beam search** | 46.50/22.48/43.66 | 47.82/24.65/39.67 | 31.04/9.96/24.37 | 54.66/30.27/45.46 | 3.70% |
| diverse beam | 46.31/22.48/43.47 | 47.79/24.53/39.51 | 31.00/9.95/24.08 | 54.57/29.67/45.55 | 3.26% |
| nucleus | 46.45/22.46/43.54 | 47.67/24.50/39.47 | 31.09/10.01/24.31 | 54.61/30.04/45.63 | 3.51% |
| *calibration checkpoint selection* | | | | | |
| ROUGE | 46.66/22.66/43.84 | 48.03/24.78/39.79 | 30.94/9.98/24.43 | 54.63/30.03/45.79 | 3.96% |
| **perplexity** | 47.36/24.02/44.45 | 47.96/24.74/39.78 | 31.04/10.08/24.53 | 54.65/30.11/46.00 | 4.93% |

**Similarity Function** We compare our proposed similarity function, using models' latent states at decoder output representation $s_\theta(\hat{\mathbf{y}}, \bar{\mathbf{y}}; \mathbf{x})$ (subsection 2.1), to directly optimizing the evaluation metric ROUGE. They perform similarly on all datasets even when evaluation metrics are ROUGE scores. We also test a variant of our similarity function by replacing decoder representation $emb(\mathbf{y}, \mathbf{x})$ with

token embeddings. This variant has lower performance, which suggests benefits of contextualized and input-dependent representations.

**Calibration Loss** Calibrated models with all loss types significantly improve over fine-tuned-only models. Rank loss performs the best followed by margin, list rank and then reward. Reward maximization has the advantage of no hyper-parameters $\beta$ (Equation 1) to sweep while rank and margin loss have smaller training memory footprints. Rank loss showing the best gain indicates that relative ordering of candidates is more important than the absolute value of their similarity to the target.

**Regularization Loss** Cross entropy and KL divergence regularization perform similarly. About 85% of the calibration gain remains if regularization is removed.

**Calibration Candidates Decoding Method** We choose hyper-parameters for calibration candidates decoding methods based on validation set. The optimal decoding method is dataset dependent, however the differences between methods are small and the worst method achieves 90% of the gains of the best one. Beam search yields the highest average quality. This is opposite to the findings in the two-stage reranking systems (Liu and Liu, 2021; Ravaut et al., 2022b; Liu et al., 2022), where more diverse decoding strategies are preferred.

**Checkpoint Selection for Fine-tuned Model** We compare ROUGE-selected and perplexity-selected checkpoints. The experiments show that starting calibration from the perplexity-selected checkpoint yields same or better performance with the biggest gap on CNN/DailyMail dataset.

**TL;DR:** We recommend a simple recipe: select the fine-tuned model's checkpoint by its validation set perplexity; decode candidates using beam search; calibrate the model with rank loss (based on decoder states similarities) and KL divergence regularization.

### 3.4 BENEFITS OF CALIBRATED SEQUENCE LIKELIHOOD

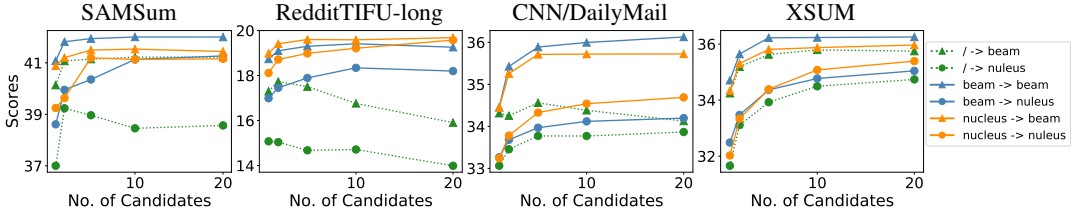

Figure 2: Effect of decoding methods on calibrated and fine-tuned only models. Colors indicate calibration method. Markers indicate evaluation decoding method. Hyper-parameters at Appendix F.

**Calibrated models' quality monotonically improves as the number of decoding candidates increase,**[1] regardless of the calibration-decoding and evaluation-decoding methods, as shown in Figure 2. On the other hand, fine-tuned-only models suffer from decreased quality when the number of decodes exceeds an optimal value. Once a model is calibrated with either decoding method, it performs well with both at evaluation time. Decoding with beam search yields higher scores, verified up to 20 decodes. When the calibration-decoding and the evaluation-decoding method align, the final quality is slightly better than the mismatched settings. CNN/DailyMail, XSUM, and SAMSum datasets work best with beam search, however RedditTIFU-long works better with nucleus sampling and decoding it with a larger number of candidates may achieve better results.

**Calibrated models do not require length normalization.** As shown in Table 2, length normalization (commonly implemented as $\alpha$ for beam search) is essential for fine-tuned-only models which bias towards longer sequences at decoding time. In contrast, length normalization has minimal effect on calibrated models.

**Calibrated models suffer from far fewer repetitions.** The repetition rate (rep%) measures a common mode of model failures. It is defined as the percentage of examples that contain any kind of consecutive repeated word n-grams, While length normalization helps general quality on the fine-tuned-only models, it leads to a side-effect of higher repetitions. Calibrated models, with or without

---

[1]At evaluation-decoding time, the candidate with the highest sequence probability is selected to compute quality for both beam search and nucleus sampling.

Table 2: Comparison between fine-tuned only models and calibrated models with or w/o brevity penalty $\alpha$ on overall quality (R1 / R2 / RL) and repetitions' occurrence percentage (rep%). Hyper-parameters at Appendix G.

| SLiC | $\alpha$ | CNN/DailyMail | | XSUM | | RedditTIFU-long | | SAMSum | | $\Delta$ |
|------|----------|---------------|------|------|------|-----------------|------|--------|------|----------|
| | | R1 / R2 / RL | rep% | R1 / R2 / RL | rep% | R1 / R2 / RL | rep% | R1 / R2 / RL | rep% | avg |
| gold reference | | - | 0.03 | - | 0.01 | - | 0.09 | - | 0.05 | |
| ✗ | ✗ | 39.37/19.67/36.89 | 0.03 | 46.96/24.29/39.19 | 0.03 | 26.62/8.91/21.77 | 0.26 | 50.28/27.25/42.69 | 0.00 | -5.15% |
| ✗ | ✓ | 44.74/21.83/41.92 | 0.13 | 47.23/24.31/39.12 | 0.07 | 26.84/9.08/21.92 | 0.90 | 53.67/29.35/44.75 | 0.20 | 0.00% |
| ✓ | ✗ | 46.44/22.38/43.57 | 0.02 | 47.57/24.42/39.46 | 0.03 | 30.99/9.95/24.39 | 0.03 | 54.42/29.98/45.56 | 0.00 | 3.31% |
| ✓ | ✓ | 46.49/22.55/43.63 | 0.03 | 47.77/24.48/39.49 | 0.03 | 30.98/9.96/24.30 | 0.12 | 54.64/30.01/45.17 | 0.08 | 3.42% |

length normalization, have a much lower repetition rate. When we compare with the repetition rate in the gold reference (repetition may occur naturally), calibrated models without length normalization have similar or lower repetition rate.

**TL;DR:** Calibrated models do not require decoding heuristics such as beam size optimization, length normalization and repetition blocking.

## 3.5 Scaling Properties of Calibrated Models

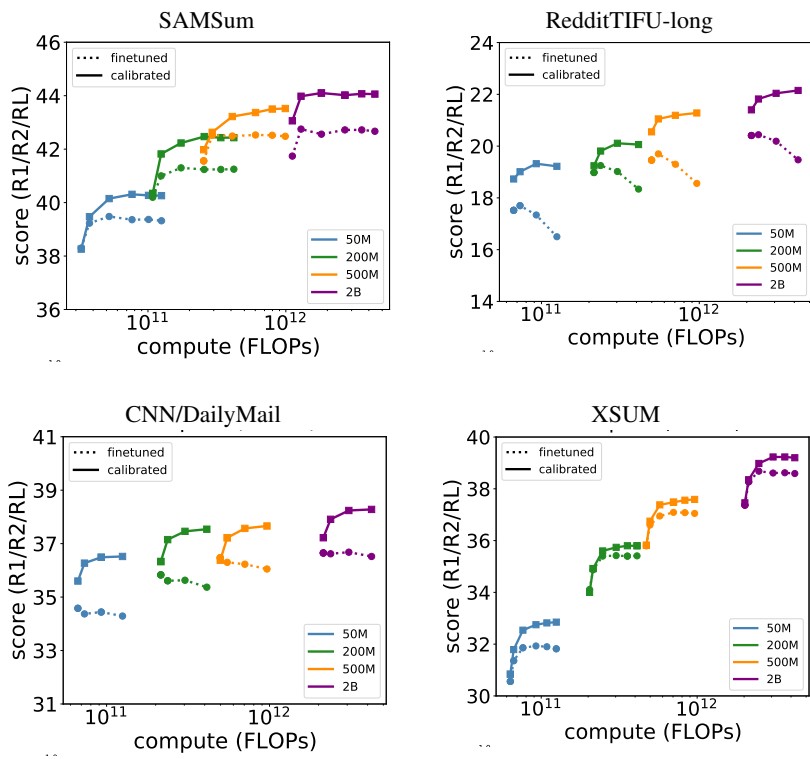

Figure 3: Quality and inference compute trade-off comparison between fine-tuned only and calibrated models. Inference compute is scaled by increasing model parameters (different colors) and number of decoding candidates (dots on the same line). Hyper-parameters at Appendix I.

Scaling properties are important for projecting a technique's future relevance as models scale up (Kaplan et al., 2020a). In Figure 3, we compare generation quality versus inference compute at different model sizes and number of decoding candidates using beam search. Appendix H describes the method to estimate inference compute FLOPs.

As mentioned earlier in subsection 3.4, fine-tuned-only models have optimal decoding beam sizes while calibrated models' performance monotonically increase with larger decoding beam sizes. Even in the case of greedy decoding (beam size of 1), the calibrated models' performance exceeds the fine-tuned-only models, by a large margin for some datasets (CNN/DailyMail and RedditTIFU-long). Their gaps grow larger with increasing number of beam sizes.

**The magnitude of quality improvement from calibration persists over models sizes spanning from 50M to 2B**. There is no obvious sign of diminishing return as model size scales up.

**Inference compute may be used for decoding rather than on larger models.** A calibrated model, once trained, can improve its performance by decoding more candidates, usually more effectively in the beginning, although returns diminish over 10 candidates. In some cases (SAMSum and especially CNN/DailyMail), a smaller model decoding more candidates can beat a larger one at both quality and efficiency.

**TL;DR:** Calibration benefits persist as model sizes scale up. Smaller calibrated models can outperform larger ones under the same inference compute budget.

## 3.6 FINAL RESULTS

Table 3: Calibrated PEGASUS$_{2B}$ comparing with prior SOTA results: BRIO[a](Liu et al., 2022), ULL[b](Tay et al., 2022), ST-MoE[c](Zoph et al., 2022), UniLMv2[d](Bao et al., 2020), Masque[e](Nishida et al., 2019), and BART+R3F[f](Aghajanyan et al., 2021). † is on validation set. * is on unknown split. See hyper-parameters in Appendix J.

| Dataset | Prior SOTA | | Our fine-tuned (2B) | Our calibrated (2B) |
|---|---|---|---|---|
| | #params | R1 / R2 / RL | R1 / R2 / RL | R1 / R2 / RL |
| CNN/DailyMail | 340M [a] | 47.78/23.55/44.57 | 44.31/21.91/41.41 | 47.97/24.18/44.88 |
| XSUM | 268B [c] | – /27.1/– | 49.57/26.77/41.41 | 49.77/27.09/42.08 |
| RedditTIFU-long | 340M [f] | 30.31/10.98/24.74* | 28.73/10.12/23.24 | 32.03/11.13/25.51 |
| SAMSum | 20B [b] | –/29.60/– | 53.64/29.21/44.83 | 54.37/29.88/45.89 |
| SQuAD QG | 110M [d] | –/–/52.13 | –/–/52.59 | –/–/53.28 |
| MSMARCO NLG † | UNK[e] | –/–/69.77 | –/–/70.73 | –/–/71.06 |
| WebNLG-en | 20B [b] | –/55.40/– | 76.96/52.97/62.56 | 78.09/55.52/65.06 |
| CommonGen † | 20B [b] | –/37.40/– | 66.49/36.17/58.82 | 68.95/38.49/60.13 |

We calibrate the fine-tuned PEGASUS$_{2B}$ models on 8 language generation tasks using the simple recipe identified in subsection 3.3 and evaluate them with beam search without decoding heuristics (subsection 3.4). The only hyper-parameter we optimize for SLiC is learning rate $lr$ (Appendix J). We use beam size 5 for fine-tuned-only models and 10 for calibrated models.

As shown in Table 3, calibrated models show consistent improvement over fine-tuned-only models across datasets and tasks. Overall, our calibrated models exceed or match the SOTA models on all datasets. On XSUM, SAMSum, WebNLG-en and CommonGen, our calibrated 2B models are ten to a hundred times smaller than the SOTA models.

**TL;DR:** PEGASUS$_{2B}$ achieves SOTA results on a wide range of language generation tasks using a simple SLiC recipe while eliminating decoding heuristics.

## 4 RELATED WORKS

In classification, model calibration often refers to matching output probabilities with expected accuracy. In our case of sequence generation, how to generalize this notion is unclear. Kuleshov and Liang (2015) explores generalizing probabilistic calibration to structured prediction, but we only focus on aligning the sequence likelihood with target sequence similarity. Other approaches in this vein are described below.

### 4.1 RL APPROACHES

Paulus et al. (2018) directly optimizes evaluation metric ROUGE in RL fine-tuning stage. One issue is that ROUGE metric does not enforce fluency. The authors found summaries to be not always readable and proposed that using a mixed training objective works better.

Ziegler et al. (2019); Stiennon et al. (2020) collects human judgements on fine-tuned models' decodes to train a reward model that ranks candidates according to human preferences. The supervised policy is then fine-tuned against the reward model using PPO. The authors found that optimizing their reward model results in better quality summaries than directly optimizing ROUGE.

## 4.2 Two-stage reranking approaches

SimCLS (Liu and Liu, 2021) proposes formulating text generation as a reference-free quality estimation problem assisted by contrastive learning. The first stage decodes candidates with diverse beam search and a RoBERTa based model is used to rank them in the second stage.

SummaReRanker (Ravaut et al., 2022a) observes improved performance when training the generation and the reranking models on two non-overlapping halves of the fine-tuning data compared to training two models on the same data.

Bhattacharyya et al. (2021) trains an energy-based model to mimic the behavior of the task measure such as BLEU scores.

Lee et al. (2021); Fernandes et al. (2022) train rerankers for neural machine translation (NMT) that predicts the observed distribution of desired automatic metrics (BLEU, COMET and BLEURT) over the n-best list.

BRIO (Liu et al., 2022) includes a two-stage reranking system that uses sequence-to-sequence generation models. It is shown that the sequence-to-sequence reranker has better performance than encoder-only models in providing ranking scores.

## 4.3 Multi Task Learning with Sequence-level Loss

Edunov et al. (2018) surveys a range of classical objective functions for structured prediction and apply them to sequence-to-sequence models. Their experiments showed that combining sequence-level objectives with token-level objectives yields improved performance on translation and summarization datasets.

Sun and Li (2021) combines contrastive learning objective with negative log-likelihood to decrease the likelihood of the model generated "silver" summaries meanwhile increasing the likelihood of the "gold" references.

Wieting et al. (2019) introduces an alternative reward function for optimizing neural machine translation systems that is based on semantic similarity.

BRIO (Liu et al., 2022) demonstrates that multi task learning of sequence candidates with contrastive reranking and token-level generation has better performance compared to a two-stage reranking system. The ranking order is determined by similarity to target using external metrics (ROUGE, BERTScore). Models trained to rank by ROUGE also perform well measured on BERTScore and vice versa.

Lukasik et al. (2020) extends label smoothing from classification tasks to semantic label smoothing for sequence-to-sequence learning. Their technique adds sequence-level losses that smooth over well-formed relevant sequences that are similar to the target sequence semantically and on n-gram level.

## 5 Conclusion

We propose adding a third stage of sequence likelihood calibration (SLiC) after the pretraining and fine-tuning stages for conditional language generation. A simple yet effective recipe for SLiC is using decoder-state similarity, selecting the fine-tuned model's checkpoint by perplexity, decoding candidates with beam search, calibrating with rank loss and KL divergence regularization. We are able to eliminate all decoding heuristics for calibrated models. The benefits of calibration persist as models scale up in size. Smaller calibrated models might outperform larger ones under the same inference compute budget. By calibrating a PEGASUS$_{2B}$ model, we exceed or match state-of-the-art results on 8 datasets spanning abstractive summarization, generative question answering, question generation and data-to-text tasks.

In this work we focus on the setting where ground-truth output sequences are provided. However, this presupposes high-quality labels that are not always available. In the future, we plan to extend SLiC to general language modeling and explore more types of latent similarities.

ACKNOWLEDGEMENT

We thank David Grangier for early and engaging discussions, and Noah Fiedel for feedback on the paper.

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

# A  DATASETS PROPERTIES

## A.1  DATASETS AND TASKS

**CNN/DailyMail** (Hermann et al., 2015; See et al., 2017) summarization dataset contains 313k articles from the CNN and Daily Mail newspapers with bullet point summaries. The summaries are on average 3-4 sentences and relatively extractive.[2]

**XSUM** (Narayan et al., 2018) summarization dataset consists of 227k BBC articles from 2010 to 2017 with a single sentence highly abstractive summary. Sometimes the summary contains information not present in the article.[3]

**RedditTIFU-long** (Kim et al., 2019) summarization dataset contains 42k posts of informal stories from sub-reddit TIFU from 2013-Jan to 2018-Mar with author written summaries. The style and length of the summaries are very diverse.[4]

**SAMSum** (Gliwa et al., 2019) summarization dataset contains 16k high-quality chat-dialogues and their summaries written by linguists.[5]

**SQuAD QG** (Zhou et al., 2017; Du et al., 2017) is the task of generating a question from a passage-answer pair extracted from the SQuAD dataset (Rajpurkar et al., 2016). In particular, we use the split of Du et al. (2017), consisting of 75,722, 10,570, and 11,877 examples for training, validation, and testing, respectively.[6]

**MSMARCO NLG** (Bajaj et al., 2016) is a large scale dataset focused on machine reading comprehension and question answering. The original QA dataset consists of 1,010,916 queries. However, we work on the NLGEN data that is a subset of the QA data consisting of 182,669 queries, each with a well formed answer. The task is to generate a well formed answer to an input query and a set of answering passages.[7]

**WebNLG-en** (Gardent et al., 2017) consists of 16,095 data inputs in the form of sets of RDF triples extracted from DBpedia. Each data point was verbalized by humans in more-than-one natural texts, leading to a total of 38,872 data-text pairs. [8]

**CommonGen** (Lin et al., 2020) introduces a task of generating a coherent sentence describing an input set of common concepts. The dataset consists of a total of 35,141 common concept sets, split into 32,651/993/1,497 training/validation/test sets. There are 67,389, 4,018 and 6,042 sentences in training, validation and test, respectively.[9]

Table 4: Statistics of datasets.

| dataset | # of examples train/val/test | avg words input | avg words target | extractiveness coverage | extractiveness density |
|---|---|---|---|---|---|
| CNN/DailyMail | 287K / 13K / 11K | 698.60 | 49.53 | 87.8% | 3.77 |
| XSUM | 203K / 11K / 11K | 383.17 | 21.74 | 63.9% | 1.06 |
| RedditTIFU-long | 34K / 4K / 4K | 396.15 | 21.02 | 68.4% | 1.27 |
| SAMSum | 14,732 / 818 / 819 | 97.23 | 21.00 | 68.0% | 1.46 |
| SQuAD QG | 76K / 11K / 12K | 128.72 | 10.24 | 64.7% | 1.63 |
| MSMARCO NLG | 152K / 12K / 12K | 588.50 | 14.07 | 97.5% | 7.78 |
| WebNLG-en | 35K / 1667 / 1779 | 17.50 | 20.51 | 48.7% | 1.3 |
| CommonGen | 67K / 993 / 1497 | 3.27 | 10.10 | 22.0% | 0.22 |

---

[2]https://www.tensorflow.org/datasets/catalog/cnn_dailymail
[3]https://www.tensorflow.org/datasets/catalog/xsum
[4]https://www.tensorflow.org/datasets/catalog/reddit_tifu
[5]https://www.tensorflow.org/datasets/catalog/samsum
[6]https://www.tensorflow.org/datasets/catalog/squad_question_generation
[7]https://huggingface.co/datasets/ms_marco/viewer/v2.1
[8]https://www.tensorflow.org/datasets/catalog/gem#gemweb_nlg_en
[9]https://www.tensorflow.org/datasets/catalog/gem#gemcommon_gen_default_config

## B    MODEL ARCHITECTURE

Model sizes and their configurations are reported in Table 5.

Table 5: Model sizes.

| name | num layers enc/dec | hidden size | num heads | MLP size | # num params excluding embs | # total |
|---|---|---|---|---|---|---|
| PEGASUS$_{SMALL}$ | 8/8 | 512 | 8 | 1024 | 49M | 108M |
| PEGASUS$_{BASE}$ | 12/12 | 768 | 12 | 3072 | 198M | 272M |
| PEGASUS$_{LARGE}$ | 16/16 | 1024 | 16 | 4096 | 470M | 568M |
| PEGASUS$_{2B}$ | 24/24 | 1024 | 16 | 16384 | 1913M | 2012M |

## C    HYPER-PARAMETER NOTATIONS

Table 6: Summary of hyper-parameters notations.

| notation symbol | meaning |
|---|---|
| lr | learning rate |
| $\alpha$ | length penalty factor in beam search |
| $\beta$ | scale of ranking margin in calibration loss (Equation 1) |
| $\lambda$ | relative scale of regularization loss compared with calibration loss (section 2) |

## D    EVALUATION SCRIPTS

For summarization tasks, we use pypi package `rouge-score` to report ROUGE numbers. We report `rougeLsum` for ROUGE-L.

For SQuAD QG and MSMARCO NLG, we use the original evaluation scripts provided by Du et al. (2017) and Bajaj et al. (2016), respectively. For WebNLG-en and CommonGen, we use the versions from the GEM benchmark (Gehrmann et al., 2021) and report using the GEM evaluation framework. Those scripts mainly differ in text tokenization methods.

# E ABLATION STUDY

SLiC methods for ablation study are reported in Table 7.

Table 7: Experimental settings for ablation studies.

| Ablation | | calibration | | | | | | evaluation |
|---|---|---|---|---|---|---|---|---|
| | decoding | sim fn | loss | regularization | ckpt | extra | | decoding |
| fine-tuned | - | - | - | - | - | - | | beam 5 |
| similarity function | | | | | | | | |
| ROUGE | beam 15 | ROUGE | reward | cross entropy | ROUGE | fix lr | | beam 5 |
| decoder repr | beam 15 | $s_\theta(\mathbf{y}, \hat{\mathbf{y}}, \mathbf{x})$ | reward | cross entropy | ROUGE | fix lr | | beam 5 |
| token emb | beam 15 | $s_{tok}(\mathbf{y}, \hat{\mathbf{y}})$ | reward | cross entropy | ROUGE | fix lr | | beam 5 |
| calibration loss | | | | | | | | |
| rank | beam 15 | $s_\theta(\mathbf{y}, \hat{\mathbf{y}}, \mathbf{x})$ | rank | cross entropy | ROUGE | best lr, $\beta$ | | beam 5 |
| margin | beam 15 | $s_\theta(\mathbf{y}, \hat{\mathbf{y}}, \mathbf{x})$ | margin | cross entropy | ROUGE | best lr, $\beta$ | | beam 5 |
| list rank | beam 15 | $s_\theta(\mathbf{y}, \hat{\mathbf{y}}, \mathbf{x})$ | list rank | cross entropy | ROUGE | best lr, $\beta$ | | beam 5 |
| reward | beam 15 | $s_\theta(\mathbf{y}, \hat{\mathbf{y}}, \mathbf{x})$ | reward | cross entropy | ROUGE | best lr, $\beta$ | | beam 5 |
| regularization loss | | | | | | | | |
| none | beam 15 | $s_\theta(\mathbf{y}, \hat{\mathbf{y}}, \mathbf{x})$ | rank | - | ROUGE | fix lr, $\beta$ | | beam 5 |
| cross entropy | beam 15 | $s_\theta(\mathbf{y}, \hat{\mathbf{y}}, \mathbf{x})$ | rank | cross entropy | ROUGE | fix lr, $\beta$ | | beam 5 |
| KL divergence | beam 15 | $s_\theta(\mathbf{y}, \hat{\mathbf{y}}, \mathbf{x})$ | rank | KL divergence | ROUGE | fix lr, $\beta$ | | beam 5 |
| calibration decoding method | | | | | | | | |
| beam | beam 15 | $s_\theta(\mathbf{y}, \hat{\mathbf{y}}, \mathbf{x})$ | reward | cross entropy | ROUGE | fix lr | | beam 5 |
| diverse beam | diverse beam 15 | $s_\theta(\mathbf{y}, \hat{\mathbf{y}}, \mathbf{x})$ | reward | cross entropy | ROUGE | fix lr | | beam 5 |
| nucleus | nucleus 15 | $s_\theta(\mathbf{y}, \hat{\mathbf{y}}, \mathbf{x})$ | reward | cross entropy | ROUGE | fix lr | | beam 5 |
| calibration checkpoint selection | | | | | | | | |
| ROUGE | beam 15 | $s_\theta(\mathbf{y}, \hat{\mathbf{y}}, \mathbf{x})$ | reward | cross entropy | ROUGE | fix lr | | beam 5 |
| perplexity | beam 15 | $s_\theta(\mathbf{y}, \hat{\mathbf{y}}, \mathbf{x})$ | reward | cross entropy | perplexity | fix lr | | beam 5 |

# F  DECODING METHODS

SLiC methods for decoding calibrated models are reported in Table 8. At evaluation time, models are decoded with 1, 2, 5, 10 and 20 candidates. ROUGE numbers in Figure 2 are reported in Table 9.

Table 8: Experimental settings for calibrated models' decoding analysis.

| name | calibration | | | | | | evaluation | |
| | decoding | sim fn | loss | regularization | ckpt | extra | decoding | $\alpha$ |
| --- | --- | --- | --- | --- | --- | --- | --- | --- |
| / → beam | | | | | | | beam 1-20 | ✓ |
| / → nucleus | | | | | | | nucleus 1-20 | ✓ |
| beam → beam | beam 15 | $s_\theta(\mathbf{y}, \hat{\mathbf{y}}, \mathbf{x})$ | reward | cross entropy | ROUGE | fix lr | beam 1-20 | ✗ |
| beam → nucleus | beam 15 | $s_\theta(\mathbf{y}, \hat{\mathbf{y}}, \mathbf{x})$ | reward | cross entropy | ROUGE | fix lr | nucleus 1-20 | ✗ |
| nucleus → beam | nucleus 15 | $s_\theta(\mathbf{y}, \hat{\mathbf{y}}, \mathbf{x})$ | reward | cross entropy | ROUGE | fix lr | beam 1-20 | ✗ |
| nucleus → nucleus | nucleus 15 | $s_\theta(\mathbf{y}, \hat{\mathbf{y}}, \mathbf{x})$ | reward | cross entropy | ROUGE | fix lr | nucleus 1-20 | ✗ |

Table 9: ROUGE (R1 / R2 / RL) numbers of the decoding curves.

| SLiC → decoding | num decodes | CNN/DailyMail R1 / R2 / RL | XSUM R1 / R2 / RL | RedditTIFU-long R1 / R2 / RL | SAMSum R1 / R2 / RL |
| --- | --- | --- | --- | --- | --- |
| / → beam | 1 | 45.11/21.15/42.34 | 46.18/22.84/38.07 | 27.78/8.40/22.18 | 52.86/27.89/43.85 |
| | 2 | 44.54/21.62/41.73 | 46.94/23.87/38.89 | 27.75/8.98/22.37 | 53.42/29.11/44.56 |
| | 5 | 44.78/21.99/41.93 | 47.26/24.38/39.24 | 26.88/9.09/21.95 | 53.47/29.25/44.53 |
| | 10 | 44.58/21.86/41.71 | 47.29/24.60/39.41 | 25.51/8.78/21.04 | 53.70/29.22/44.63 |
| | 20 | 44.33/21.64/41.43 | 47.13/24.62/39.36 | 24.10/8.32/20.06 | 53.74/29.21/44.64 |
| / → nucleus | 1 | 44.09/19.88/41.24 | 43.76/20.42/35.51 | 25.33/6.84/19.78 | 50.51/24.56/40.85 |
| | 2 | 44.31/20.36/41.50 | 45.03/21.80/36.96 | 24.82/6.95/19.72 | 52.17/26.91/43.02 |
| | 5 | 44.43/20.81/41.67 | 45.61/22.63/37.83 | 23.80/6.84/19.43 | 51.50/26.70/43.02 |
| | 10 | 44.28/20.94/41.54 | 46.06/23.22/38.37 | 23.45/6.98/19.44 | 50.69/26.28/42.70 |
| | 20 | 44.25/21.13/41.53 | 46.06/23.57/38.62 | 21.87/6.81/18.39 | 50.53/26.58/42.72 |
| beam → beam | 1 | 45.72/20.87/42.89 | 46.71/23.16/38.65 | 30.00/9.20/23.82 | 54.24/28.67/44.59 |
| | 2 | 46.46/21.96/43.58 | 47.46/24.17/39.47 | 30.24/9.56/24.11 | 54.68/29.71/45.02 |
| | 5 | 46.72/22.55/43.87 | 47.88/24.79/40.05 | 30.25/9.80/24.26 | 54.78/29.75/45.27 |
| | 10 | 46.81/22.67/43.95 | 47.83/24.82/40.06 | 30.31/9.89/24.39 | 54.63/30.01/45.20 |
| | 20 | 46.90/22.83/44.04 | 47.83/24.86/40.07 | 30.02/9.80/24.29 | 54.74/29.98/45.15 |
| beam → nucleus | 1 | 44.83/19.59/41.92 | 44.73/20.99/36.52 | 28.19/7.89/22.05 | 52.26/26.19/42.07 |
| | 2 | 45.16/20.01/42.28 | 45.55/21.92/37.56 | 28.66/8.22/22.58 | 53.15/27.61/43.45 |
| | 5 | 45.35/20.34/42.49 | 46.15/22.87/38.45 | 28.83/8.62/23.06 | 53.50/27.80/44.18 |
| | 10 | 45.46/20.51/42.59 | 46.39/23.33/38.85 | 28.90/9.10/23.47 | 53.99/28.71/44.89 |
| | 20 | 45.46/20.63/42.63 | 46.53/23.67/39.07 | 28.60/9.01/23.39 | 54.22/28.68/45.19 |
| nucleus → beam | 1 | 45.66/20.93/42.77 | 46.50/22.93/37.97 | 30.57/9.45/23.68 | 53.81/28.71/44.23 |
| | 2 | 46.19/21.91/43.29 | 47.29/23.93/38.90 | 30.94/9.82/24.06 | 53.99/29.25/44.30 |
| | 5 | 46.47/22.50/43.56 | 47.74/24.43/39.36 | 31.10/10.00/24.22 | 54.29/29.49/44.62 |
| | 10 | 46.39/22.57/43.52 | 47.78/24.52/39.41 | 31.02/10.00/24.22 | 54.25/29.54/44.70 |
| | 20 | 46.34/22.63/43.48 | 47.83/24.63/39.49 | 31.11/10.09/24.29 | 54.17/29.46/44.59 |
| nucleus → nucleus | 1 | 44.68/19.69/41.75 | 44.35/20.69/35.80 | 29.85/8.68/22.94 | 52.55/26.98/42.63 |
| | 2 | 45.14/20.24/42.20 | 45.50/21.94/37.13 | 30.31/9.22/23.45 | 52.97/27.33/43.00 |
| | 5 | 45.58/20.81/42.65 | 46.43/22.93/38.19 | 30.46/9.44/23.81 | 54.10/28.86/44.77 |
| | 10 | 45.73/21.05/42.82 | 46.91/23.65/38.90 | 30.69/9.58/24.11 | 54.02/28.82/44.70 |
| | 20 | 45.78/21.26/42.88 | 47.19/24.00/39.14 | 31.04/9.89/24.44 | 53.78/29.02/44.66 |

## G  LENGTH NORMALIZATION

Experimental settings for length normalization analysis is reported in Table 10. Brevity penalty $\alpha$ is chosen as the best value for fine-tuned models' ROUGE performance on validation dataset or disabled.

Table 10: Experimental settings for length normalization study.

| SLiC | $\alpha$ | | | calibration | | | | evaluation | |
|---|---|---|---|---|---|---|---|---|---|
| | | decoding | sim fn | loss | regularization | ckpt | extra | decoding | $\alpha$ |
| ✗ | ✗ | | | | | | | beam 5 | ✗ |
| ✗ | ✓ | | | | | | | beam 5 | ✓ |
| ✓ | ✗ | beam 15 | $s_\theta(\mathbf{y}, \hat{\mathbf{y}}, \mathbf{x})$ | best | cross entropy | ROUGE | best lr, $\beta$ | beam 5 | ✗ |
| ✓ | ✓ | beam 15 | $s_\theta(\mathbf{y}, \hat{\mathbf{y}}, \mathbf{x})$ | best | cross entropy | ROUGE | best lr, $\beta$ | beam 5 | ✓ |

## H  MODEL FLOPS ESTIMATION

We extends formulations in Table 1 of Kaplan et al. (2020b) to estimate FLOPs of our transformer encoder decoder models following the formula:

$$
\begin{aligned}
total\_C &= C_{enc} \times n_{enc-ctx} + C_{dec} \times n_{dec-ctx} \times m \\
C_{enc} &= 2N_{enc} + 2n_{enc-layer}n_{enc-ctx}d_{enc-attn} \\
C_{dec} &= 2N_{dec} + n_{dec-layer}n_{dec-ctx}d_{dec-attn}
\end{aligned}
\tag{3}
$$

where $m$ is the number of decoder candidates, other notations can be referenced in Table 1 of Kaplan et al. (2020b). Because of upper triangle attention masking, the effective decoder attention context length is half of sequence lengths instead of full sequence lengths as in the encoder. Extra computation incurred by different decoding methods are omitted as they are much smaller.

## I  SCALING

SLiC method for scaling curves are reported in Table 11. At evaluation time, models are decoded with 1, 2, 5, 10, and maybe 15, 20 candidates. ROUGE numbers in Figure 3 are reported in Table 12.

Table 11: Experimental settings for scaling.

| model | | calibration | | | | | | evaluation |
|---|---|---|---|---|---|---|---|---|
| | decoding | sim fn | loss | regularization | ckpt | extra | | decoding |
| fine-tuned | | | | | | | | beam 1-20 |
| calibrated | beam 15 | $s_\theta(\mathbf{y}, \hat{\mathbf{y}}, \mathbf{x})$ | reward | cross entropy | ROUGE | best lr | | beam 1-20 |

Table 12: ROUGE (R1 / R2 / RL) numbers of the scaling curve.

| size | decodes | CNN/DailyMail R1 / R2 / RL | XSUM R1 / R2 / RL | RedditTIFU-long R1 / R2 / RL | SAMSum R1 / R2 / RL |
|------|---------|----------------------------|-------------------|------------------------------|---------------------|
| | | | fine-tuned | | |
| 50M | 1 | 43.21/19.99/40.53 | 40.91/17.80/32.98 | 25.37/6.99/20.19 | 49.78/24.45/40.67 |
| | 2 | 42.77/20.40/39.94 | 41.55/18.78/33.75 | 25.22/7.53/20.34 | 50.52/25.37/41.80 |
| | 5 | 42.92/20.45/39.96 | 41.87/19.44/34.28 | 24.41/7.61/20.00 | 50.52/25.92/42.00 |
| | 10 | 42.78/20.32/39.75 | 41.85/19.57/34.38 | 23.04/7.43/19.04 | 50.41/25.84/41.81 |
| | 15 | - | 41.79/19.59/34.31 | - | 50.46/25.89/41.77 |
| | 20 | - | 41.65/19.56/34.25 | - | 50.50/26.00/41.45 |
| 200M | 1 | 44.59/20.96/41.93 | 44.51/21.34/36.47 | 27.32/8.06/21.56 | 51.77/26.44/42.38 |
| | 2 | 44.06/21.44/41.33 | 45.24/22.22/37.15 | 27.36/8.49/21.89 | 52.35/27.40/43.27 |
| | 5 | 44.08/21.54/41.27 | 45.65/22.83/37.71 | 26.61/8.78/21.67 | 52.48/27.72/43.70 |
| | 10 | 43.84/21.30/40.96 | 45.61/22.93/37.70 | 25.80/8.37/20.85 | 52.40/27.64/43.67 |
| | 15 | - | 45.55/22.94/37.71 | - | 52.35/27.69/43.67 |
| | 20 | - | 45.54/22.99/37.71 | - | 52.38/27.68/43.68 |
| 500M | 1 | 45.34/21.47/42.60 | 46.27/23.02/38.12 | 27.79/8.42/22.18 | 53.05/27.96/43.66 |
| | 2 | 44.93/21.83/42.15 | 46.99/23.90/38.89 | 27.76/8.99/22.36 | 53.73/29.07/44.75 |
| | 5 | 44.78/21.98/41.92 | 47.26/24.37/39.23 | 26.85/9.09/21.94 | 53.94/29.01/44.53 |
| | 10 | 44.59/21.86/41.71 | 47.27/24.59/39.40 | 25.97/8.74/20.99 | 53.67/29.31/44.62 |
| | 15 | - | 47.20/24.63/39.41 | - | 53.71/29.22/44.63 |
| | 20 | - | 47.15/24.62/39.37 | - | 53.68/29.16/44.61 |
| 2B | 1 | 45.52/21.70/42.73 | 47.89/24.54/39.67 | 28.82/9.29/23.13 | 53.40/28.01/43.82 |
| | 2 | 45.37/21.95/42.54 | 48.66/25.61/40.55 | 28.60/9.60/23.12 | 53.89/29.47/44.88 |
| | 5 | 45.40/22.09/42.56 | 48.94/26.18/40.91 | 27.86/9.87/22.84 | 53.98/29.08/44.62 |
| | 10 | 45.29/21.82/42.44 | 48.91/26.08/40.84 | 27.52/9.01/21.86 | 53.95/29.61/44.61 |
| | 15 | - | 48.96/26.12/40.78 | - | 53.92/29.61/44.63 |
| | 20 | - | 48.75/26.20/40.83 | - | 53.86/29.57/44.59 |
| | | | calibrated | | |
| 50M | 1 | 44.31/20.82/41.65 | 41.41/17.95/33.15 | 27.15/7.57/21.48 | 49.85/24.62/40.33 |
| | 2 | 44.91/21.76/42.13 | 42.27/18.99/34.11 | 27.34/8.00/21.70 | 50.89/25.71/41.82 |
| | 5 | 45.12/22.10/42.25 | 42.88/19.84/34.89 | 27.49/8.43/22.02 | 51.53/26.58/42.34 |
| | 10 | 45.15/22.20/42.22 | 43.01/20.13/35.11 | 27.32/8.42/21.92 | 52.08/26.67/42.17 |
| | 15 | - | 43.13/20.15/35.16 | - | 52.04/26.66/42.10 |
| | 20 | - | 43.14/20.19/35.21 | - | 51.90/26.71/42.16 |
| 200M | 1 | 45.26/21.16/42.57 | 44.54/21.18/36.27 | 27.81/8.10/21.80 | 52.12/26.48/42.40 |
| | 2 | 45.97/22.25/43.21 | 45.47/22.12/37.18 | 28.38/8.68/22.38 | 53.29/28.24/43.92 |
| | 5 | 46.18/22.78/43.41 | 46.04/22.90/37.86 | 28.51/9.03/22.79 | 53.79/28.75/44.15 |
| | 10 | 46.26/22.88/43.47 | 46.21/22.99/38.01 | 28.35/9.07/22.78 | 54.06/28.86/44.49 |
| | 15 | - | 46.29/23.07/38.05 | - | 54.03/28.85/44.41 |
| | 20 | - | 46.28/23.09/38.03 | - | 53.99/28.90/44.39 |
| 500M | 1 | 45.55/20.85/42.76 | 46.42/22.93/38.12 | 29.29/9.10/23.26 | 53.31/28.43/44.18 |
| | 2 | 46.30/21.92/43.43 | 47.29/23.95/39.02 | 29.80/9.59/23.75 | 54.14/29.29/44.47 |
| | 5 | 46.55/22.48/43.68 | 47.88/24.62/39.62 | 29.83/9.84/23.91 | 54.61/29.95/45.10 |
| | 10 | 46.63/22.58/43.78 | 47.93/24.74/39.76 | 29.87/9.95/24.03 | 54.89/30.05/45.18 |
| | 15 | - | 48.05/24.80/39.83 | - | 54.88/30.27/45.34 |
| | 20 | - | 48.06/24.85/39.86 | - | 54.87/30.31/45.39 |
| 2B | 1 | 46.29/21.92/43.47 | 48.11/24.59/39.68 | 30.20/9.86/24.15 | 54.71/29.45/45.03 |
| | 2 | 46.84/22.93/43.95 | 49.04/25.55/40.46 | 30.59/10.38/24.50 | 55.17/30.68/46.09 |
| | 5 | 47.08/23.45/44.19 | 49.56/26.31/41.08 | 30.65/10.70/24.76 | 55.46/30.71/46.11 |
| | 10 | 47.08/23.57/44.19 | 49.79/26.56/41.32 | 30.75/10.79/24.91 | 55.47/30.60/46.00 |
| | 15 | - | 49.79/26.55/41.35 | - | 55.41/30.63/46.15 |
| | 20 | - | 49.76/26.54/41.30 | - | 55.38/30.65/46.14 |

## J  FINAL RESULTS

SLiC method for final results is reported in Table 13. We choose the SLiC best based on subsection 3.3. There are in total 3 hyper-parameters: learning rate $lr$ (Algorithm 1), ranking constant $\beta$ (Equation 1), and regularization strength $\lambda$ (Equation 2). We fix two of the them: $\beta$ is set to 10, and $lr * \lambda$ is set to $1e-5$. Best learning rate $lr$ is determined with hyper-parameter tuning on validation set and reported in Table 14.

Table 13: Experimental settings for length normalization study.

| model | | | calibration | | | | evaluation |
|---|---|---|---|---|---|---|---|
| | decoding | sim fn | loss | regularization | ckpt | extra | decoding |
| fine-tuned | | | | | | | beam 5 |
| calibrated | beam 15 | $s_\theta(\mathbf{y}, \hat{\mathbf{y}}, \mathbf{x})$ | rank | KL divergence | perplexity | best lr | beam 10 |

Table 14: Learning rate of final results.

| | CNN/DailyMail | XSUM | RedditTIFU-long | SAMSum |
|---|---|---|---|---|
| lr | $10^{-5}$ | $10^{-5}$ | $10^{-5}$ | $10^{-6}$ |
| | MSMARCO NLG | SQuAD QG | WebNLG-en | CommonGen |
| lr | $3 \times 10^{-6}$ | $10^{-5}$ | $10^{-6}$ | $10^{-5}$ |

