# OpenReview forum: "Calibrating Sequence likelihood Improves Conditional Language Generation"
_ICLR.cc/2023/Conference — ICLR 2023 poster_

### Official Review · Reviewer_JpRY · 2022-10-18

**Confidence:** 2
**Correctness:** 3
**Technical Novelty And Significance:** 2
**Empirical Novelty And Significance:** 3
**Recommendation:** 8

**Clarity, Quality, Novelty And Reproducibility:**

**Clarity.** The paper is well organized and clear.

**Quality.** See Strengths/Weakness section above.

**Novelty.** The method is related to the work of Liu et al. (2022) but the proposed similarity function appears to be novel.

**Reproducibility.** I believe there is no supplementary material/code. Are you planning to release the code?


**Strength And Weaknesses:**

**Strengths**

* The motivation is clear, the paper is well written and is easy to follow.
* The proposed method does not directly optimize an evaluation metric (e.g., ROUGE), and thus it’s more likely to actually improve quality rather than exploiting fragilities in the evaluation metrics.
* SLiC seems to work well without beam size optimization and length normalization.

**Weaknesses**

* I’m assuming that you’re using your proposed similarity function for all the experiments, excluding the ablation studies in $\S3.3$, where you also experiment with ROUGE. First, I think you should clarify that in the paper. Since the ablation studies appear before everything else, the “simple recipe” of that section could include the similarity function to use. Second, Table 1 suggests similar performance for both ROUGE and your method, and thus I’d be interested in seeing a more extensive comparison of these approaches.

**Minor comments/suggestions**

* For neural machine translation, specifically, there are several works proposing alternative decoding methods that try to generate high quality outputs, related to the approach of Lee et al. (2021), discussed in $\S4.2$, which I think are relevant for your work. Some of them are [1], [2], [3], and [4]. Besides, related to the work of Edunov et al. (2018), discussed in $\S4.3$, is [5] and [6].
* In $\S2.2.$, you should explicitly say which one “is the contrastive loss used in BRIO (Liu et al., 2022).”

**Other questions**

* Is there any reason to vary the hyperparameters between groups in Table 1? In particular, I see in App. C that the loss you used is different (sometimes *reward*, sometimes *rank*).
* How does your method compare to common approaches for conditional language generation in terms of memory consumption and training time? The paper would benefit from such analysis.

[1] Is MAP Decoding All You Need? The Inadequacy of the Mode in Neural Machine Translation, Eikema and Aziz, COLING 2020

[2] Energy-Based Reranking: Improving Neural Machine Translation Using Energy-Based Models, Sumanta Bhattacharyya et al., ACL 2021

[3] Quality-Aware Decoding for Neural Machine Translation, Fernandes et al., NAACL 2022

[4] High Quality Rather than High Model Probability: Minimum Bayes Risk Decoding with Neural Metrics, Freitag et al., TACL 2022

[5] Beyond BLEU: training neural machine translation with semantic similarity, Wieting et al., ACL 2019

[6] Revisiting the Weaknesses of Reinforcement Learning for Neural Machine Translation, Kiegeland and Kreutzer, NAACL 2021.



**Summary Of The Paper:**

This paper proposes sequence likelihood calibration (SLiC) for conditional language generation. Their proposal includes a first stage where candidates are generated from a fine-tuned model, and a second one, where they calibrate the previous model by continuing training it with a new objective that aims to align candidates’ sequence likelihoods by similarity to the target sequence in the model’s latent space. They perform experiments on abstractive summarization, question answering, and data-to-text generation tasks.

**Summary Of The Review:**

The paper addresses an important issue with conditional language generation models, that of uncalibrated sequence likelihood: for a given context, they are often trained with a single target sequence, and thus they lack supervision to compare (high probability) plausible hypotheses. The proposed method is interesting and the authors perform experiments that validate its applicability on different conditional language generation tasks.

---

> ### Author Response · Authors · 2022-11-16
> **Thanks for the valuable feedback. Responses in-line.**
>
> “I’m assuming that you’re using your proposed similarity function for all the experiments, excluding the ablation studies” …   the “simple recipe” of that section could include the similarity function to use”
>
> Yes, we use decoder state similarity functions for all experiments excluding the ablation studies. We added that in the “simple recipe” summary in that section as suggested.
>
> “For neural machine translation, specifically, there are several works proposing alternative decoding methods that try to generate high quality outputs, related to the approach of Lee et al. (2021), discussed in §4.2, which I think are relevant for your work. Some of them are [1], [2], [3], and [4]. Besides, related to the work of Edunov et al. (2018), discussed in §4.3, is [5] and [6].”
>
> Thanks for suggestions on related works, we updated the paper to include [2], [3], [5] which proposed training time modifications. [1] and [4] only propose alternate decoding strategies to beam search as opposed to other ranking based related works that we have mentioned. [6] seems to be a rebuttal paper to Choshen et al. (https://arxiv.org/abs/1907.01752) which suggests weaknesses of RL for NMT; that discussion may be out of scope for this paper.
>
> “you should explicitly say which one “is the contrastive loss used in BRIO (Liu et al., 2022).””
>
> We changed it to “List-wise rank loss is the contrastive loss used in BRIO (Liu et al., 2022)”
>
> “Is there any reason to vary the hyperparameters between groups in Table 1?”
>
> Some ablation experiments are conducted sequentially while others are in parallel, choices of common hyperparameters are results of what we found to be best in the previous iteration and engineering constraints on compute and memory footprint.
>
> “How does your method compare to common approaches for conditional language generation in terms of memory consumption and training time?”
>
> SLiC has similar memory/training time cost compared with two-stage-reranking and multi-task learning works in the related works section, and has much less compute compared with RL approaches.
>
> “ I believe there is no supplementary material/code. Are you planning to release the code?”
> We plan to release the model checkpoints right away and training code later through the HuggingFace library.

---

### Official Review · Reviewer_o62J · 2022-10-20

**Confidence:** 4
**Clarity, Quality, Novelty And Reproducibility:** Good
**Correctness:** 3
**Technical Novelty And Significance:** 3
**Empirical Novelty And Significance:** 3
**Recommendation:** 6

**Strength And Weaknesses:**

Strengths:

- The contributions of the SLiC are important, and do not exist in prior works. Besides proposing the SLiC, the authors also proposed a novel calibration similarity metric and used many past outstanding works as candidate decoding methods.

- Also, calibration benefits persist as model sizes scale up, so it has great potential for models plagued by former approaches which shows diminishing by scaling up. Also, small models can add calibration step to outperform larger ones. length normalization has minimal effect on calibrated models.

- Calibrated models do not require decoding heuristics and more. Only with the SLiC, the model reaches SOTA result and even better than the model with decoding heuristics, and such heuristics has minimal effect on calibrated models.

- Extensive experiments have been conducted to validate the effectiveness and relevance of the proposed method. Also, there are a lot of experiments in ablation study to get a simple and effective recipe for training.


Weaknesses:

- (1) typo in page 14, “WebNLG-en” paragraph, “data inputs in the from of sets” -> “data inputs in the form of sets” (2) typo in page 14, “CommonGen” paragraph, “introduces a tak” -> “introduces a task”

- The experimentation section mainly convinces me that this architecture is good for application. However, this work lacks of explanation on the reason of the choice of calibration loss, regularization loss, candidates decoding method, etc.

- Since this paper is highly theoretical and original. However, it suffers the problem that it does not make sense to many details in the method chapter. For example, as for the calibration loss and regularization loss, the literal part seems only notation, and lack of the reason for why taking these methods and the advantages of them. (Maybe it is included in the experiment part. However, I do think that it should also be mentioned in the method part.)

- Lack introduction of some notations. For example, I haven't found out what theta means so far. Although it is default to represent model parameters, I think it should be explained. What’s more, due to the rich content, this paper involves large amounts of parameters. Therefore, I think it is a good idea to explain the parameters in a chart. Maybe it can be appended to the appendix. In this way, the readers can follow the authors more easily.

- The description of MLE method and its related works occupied a little too much space in introduction, while the motivation to proposed method was little.

- In the section of ABLATION STUDIES OF CALIBRATION, the authors show us the improvement compared with fine-tuned models,but without the comparison between SLic and the fine-tuned model using the mentioned effective heuristics or solutions.

-  The authors propose the Calibration stage after Pre-train and Fine-tune(MLE) stage. Since fig. 3 gives the inference compute trade-off between fine-tuned only and calibrated models, the  comparison of the running time and speed between fine-tuned only and calibrated in Table 3 is lacked.

Suggestions:

- The experiment results of the proposed method should be in bold.

- The conclusion part lacks the description of the deficiencies and the prospects for future improvements.

Questions:

- There are in total 3 hyperparameters: β in calibration，lr as learning rate and λ as regularization strength. To simplify the training process, β and lr∗λ are set empirically, making the process tedious, and how the values are set in the paper?

**Summary Of The Paper:**

To tackle the issue of uncalibrated sequence likelihood, this paper proposes a new sequence likelihood calibration(SLiC) stage, which extends the common paradigm of pretraining and finetuning. The specific method is to calibrate with KL divergence and low rank after selecting the checkpoint of finetuned model by perplexity and decoding candidates with beam search.

With SLiC, decoding candidates’ quality significantly improves regardless of the decoding method, without showing any sign of diminishing returns with model scale, and former decoding heuristics will be never used any more. Furthermore, this paper proposes a novel calibration similarity metric between model decodes and targets, which is much better than the existing external metrics.

**Summary Of The Review:**

The paper proposes a novel stage called sequence likelihood calibration after the pretraining and fine-tuning stages for conditional language generation. It seems that this method has not been used in prior works and with such stage the model can outperform the SOTA results without concerning about the diminishing problems. Also the entire paper is very well-written and comprehensible, and the experimentation in this paper are quite rich.

---

> ### Author Response · Authors · 2022-11-16
> **Thanks for the valuable feedback. Responses in-line.**
>
> “typo in page 14” “Lack introduction of some notations.” “ I think it is a good idea to explain the parameters in a chart”
>
> Thanks for your suggestions, we fixed the typos and added a table of hyperparameter notations in the Appendix C. The math notations are in-line in the paper.
>
> “this work lacks of explanation on the reason of the choice of calibration loss, regularization loss, candidates decoding method”
>
> We updated the method section 2.2  to discuss the motivation of calibration loss and regularization loss more.
>
> “the authors show us the improvement compared with fine-tuned models,but without the comparison between SLic and the fine-tuned model using the mentioned effective heuristics or solutions.”
>
> Fine-tuned-only models with decoding heuristics are used as baselines in all ablation studies. One example is Table 2 where fine-tuned-only models with or without heuristics are both present. We also added one sentence at the start of the experimental section to emphasize this: “All fine-tuned-only models utilize heuristics such as beam size optimization and sweeping beam $\alpha$ for length normalization, unless specified otherwise.”
>
> “ Since fig. 3 gives the inference compute trade-off between fine-tuned only and calibrated models, the comparison of the running time and speed between fine-tuned only and calibrated in Table 3 is lacked.”
>
> Running speed depends on hardware and implementations, however given same # of parameters and # of decodes, a fine-tuned model and a calibrated model have the same running speed.
>
> “To simplify the training process, β and lr∗λ are set empirically, making the process tedious, and how the values are set in the paper?”
>
> “The experiment results of the proposed method should be in bold.”
> We bolded the names of suggested recipes in Table.1 as suggested.
>
> “The conclusion part lacks the description of the deficiencies and the prospects for future improvements.”
> In the conclusion, we added that we presuppose high-quality labeled data and a possible future direction of general language modeling.
>
> In ablation studies, we sweep \beta and lr and \lambda values, and found that \beta=10 and lr*\lambda=1e-5 works well for all tasks. In the final results (details are in appendix H), we only sweep lr around 1e-5, and anticipate this to work for new tasks.

---

### Official Review · Reviewer_YBaL · 2022-10-23

**Confidence:** 4
**Correctness:** 3
**Technical Novelty And Significance:** 3
**Empirical Novelty And Significance:** 3
**Recommendation:** 6

**Clarity, Quality, Novelty And Reproducibility:**

Regarding the presentation, although the approach itself is quite straightforward, I found it difficult to follow the narrative of the paper. At various points, such as the introduction of the calibration losses, many options are considered without proper motivation or discussion of relative merits to justify their inclusion in the experiments. I found the use of “TL;DR;” blocks at the end of certain chapters to be out-of-place; it would have been better to provide this kind of information at the beginning of the sections, ideally framed as research questions.

On related work, the approach is conceptually similar to knowledge distillation, and would benefit from comparing and contrasting both the original Hinton paper (when it comes to the KL regularization loss) and the sequence-level extension (https://arxiv.org/abs/1606.07947) when it comes to the use of sequence-level training. Some other relevant work is not cited, such as papers in non-autoregressive decoding where it is common to train on decoded outputs from an initial “teacher” model.

On reproducibility, the appendix provides many details about hyperparameter choices. On the other hand, there’s no obvious intent to release code, and some of the model sizes considered are prohibitive in terms of compute costs for many researchers.

In Figure 3, what is the “score” in the y-axis?

**Strength And Weaknesses:**

Overall, it is notable that the proposed approach yields models that do not appear to require decoding heuristics; this suggests that the calibration loss is doing something useful. Figure 3 is also fairly convincing in this respect, as it shows a distinct difference between the vanilla model and the calibrated one when further decoding samples are used (consistent improvement for the calibrated model). However, the experiments overall are not entirely convincing, since in several cases the degree of improvement over the baselines is marginal, while the proposed model is larger than the baselines in terms of # of parameters (2B). So it’s not clear that it’s the new calibration scheme that’s responsible for the improvement in these cases.

The scoring function (S2.1) is a key element of the proposal. It’s stated that “compared to using external metrics, [...] it differs from the metrics we evaluate the generation systems with and mitigates the risk of directly optimising towards imperfect metrics.” However, the proposed similarity is related to both ROUGE and BERTScore, and so it’s not clear there isn’t some bias towards better performance on the target metrics.

Finally, I would have liked to see probabilistic calibration addressed somewhere. Does the proposed calibration loss actually lead to more meaningful probability estimates? This could be measured at the token-level or on other events (cf https://papers.nips.cc/paper/2015/file/52d2752b150f9c35ccb6869cbf074e48-Paper.pdf)

**Summary Of The Paper:**

Remarking that language models can assign high probability to low “quality” outputs, this paper proposes “sequence likelihood calibration” (SLiC) as a solution. With SLiC, “decoding heuristics become unnecessary” and “quality significantly improves.” The claims are empirically supported by experiments on abstractive summarization, question generation, abstractive question answering, and data-to-text generation, with results that “exceed or match SOTA results.”

In more detail, the high-level idea proceeds in two stages. In the first stage, for each training instance, an initial model is used to sample m candidates (a variety of decoding methods are compared, including beam search and nucleus sampling). In the second stage, the initial model is further optimised using a combination of two losses:

A “calibration” loss which aims to align the sequence likelihood with the similarity to the reference target sequence. Four such losses are considered, including a ranking loss, margin loss, listwise ranking loss, and expected reward. All losses rely on a notion of similarity of a candidate sequence to the reference sequence, for which a variation of BERTScore is proposed (S2.1).
A “regularization” loss which penalizes deviations from the token-level confidences from the stage-one model. The cross-entropy to the reference labels and the KL divergence from the stage-one model predictive distribution are both considered.

Overall, beam search using the rank “calibration” loss with KL divergence “regularization” loss are found to work best.

**Summary Of The Review:**

There are some good ideas here but the presentation could be improved and the experimental results are lackluster.

---

> ### Author Response · Authors · 2022-11-16
> **Thanks for the valuable feedback. Responses in-line.**
>
> “However, the experiments overall are not entirely convincing, since in several cases the degree of improvement over the baselines is marginal, while the proposed model is larger than the baselines in terms of # of parameters (2B).”
>
> To see the effect of calibration, the most pertinent comparison is with the fine-tuned counterparts. The proposed SLiC models are not larger than the fine-tuned-only baseline models (they have the same number of parameters). We extensively show through ablations that SLiC improves significantly over fine-tuned-only models in Section 3.3, 3.4 and 3.5. Our final results in Table 3 (Section 3.6) also present a comparison between fine-tuned-only models and SLiC (with the same number of 2B parameters, see the last two columns) across different tasks. And, we observe that SLiC consistently improves over the fine-tuned-only models across tasks, and sets new SOTA performance in most tasks.
>
> The SOTA numbers are provided in Table 3 to contextualize the final performance with other works. In some cases the prior SOTA was from a significantly larger model, but in all cases there are many details that may differ. However, in the rightmost two columns everything is controlled and the effect of calibration is clearly shown.
>
> “However, the proposed similarity is related to both ROUGE and BERTScore, and so it’s not clear there isn’t some bias towards better performance on the target metrics.”
>
> Our proposed similarity function intrinsically measures the similarity of model predictions to human-written references in latent space. This is in contrast to earlier approaches which use external metrics such as  ROUGE and BERTScore as similarity measures. As a result we have two advantages: (i) we don’t rely on external models/metrics to measure similarity, and (ii) we avoid overfitting our models to target metrics (Goodhart’s Law).
>
> The proposed similarity metric is being used to calibrate our models to generate predictions that are intrinsically similar to the human-written references. Rouge and BERTScore are used to commonly evaluate the generations’ quality with respect to human reference, hence, it might be expected that our similarity metric does indeed also improve these evaluation metrics. However, the key point is that we are not directly optimizing to be better at these metrics.
>
> “Does the proposed calibration loss actually lead to more meaningful probability estimates?”
>
> In classification, there’s a well-defined and accepted notion of probabilistic calibration which is unclear how to generalize for sequence generation. Although prior work (BRIO) has shown that sequence-level learning improved token level Expected Calibration Error (ECE), in our use of the term ‘calibration’ we simply claim the sequence likelihood is aligned with target sequence similarity. [Kuleshov and Liang] is an interesting paper on calibrating structured prediction in the probabilistic sense, although our goal differs. We added this to Related Work and clarified our use of the term calibration.
>
> “many options are considered without proper motivation or discussion of relative merits to justify their inclusion in the experiments”
>
> We updated the method section 2.2  to discuss the motivation of calibration loss and regularization loss more.
>
> “the approach is conceptually similar to knowledge distillation, and would benefit from comparing and contrasting both the original Hinton paper (when it comes to the KL regularization loss) and the sequence-level extension (https://arxiv.org/abs/1606.07947) when it comes to the use of sequence-level training. “
>
> Thank you for pointing out the connection between knowledge distillation (KD) and our approach. They share the commonality that models’ output is used for training. However, it is very different in two major aspects. (1)  KD teacher outputs are used to train a different student model. In SLiC, fine-tuned-only models’ outputs are used to improve itself. (2) KD’s student training objective is to imitate the teachers’ output sequences. In SLiC, the training objective is to rank/contrast the model’s own generations..
>
> “ On the other hand, there’s no obvious intent to release code, and some of the model sizes considered are prohibitive in terms of compute costs for many researchers.”
>
> We plan to release the model checkpoints right away and training code later through huggingface.
>
> “In Figure 3, what is the “score” in the y-axis?”
>
> It is the geometric mean of ROUGE 1/2/L, we detail this in Section 3.2.

---

### Author Response · Authors · 2022-11-16
**Summary of changes in revision (November 15, 2022)**

We’d like to thank the reviewers for providing high-quality feedback on our paper. Most suggestions relate to writing and presentation and we've incorporated most.

To summarize:

- Made changes to improve clarity of the presentation.
- Added many more related work suggestions.
- On the question of code, we plan on releasing calibrated model checkpoints in an existing github repository (which we do not name to preserve anonymity) and also plan on releasing calibration training code in HuggingFace.

See below for reviewer-specific discussion points.

Please let us know if there are unresolved questions/comments that you’d like us to expand upon and we welcome further feedback.

---

### Public Comment · ~Thang_Le2 · 2023-08-31
**Question on hyperparameters and training**

Hi, thanks for the great work!

I have a question about Appendix J. In the final experiments, you mentioned that lr * lambda was set to 1e-5. What does this mean ? What was the value for lambda specifically ?

Also, I tried to reproduce the results with cross entropy as regularization on the RedditTIFU dataset (I used 16 candidates with diverse beam search, and set loss = ce_loss + list_wise_loss), but I found the ranking loss to contradict the cross entropy loss somehow (ce loss keeps increasing during training while ranking loss decreases normally) and the overall performance obtained after calibration is significantly inferior to the uncalibrated model (went from 32 R-1 to 24 R-1). Do you have any advice on this ?

And, do you have any plans on releasing the paper's code ? It would be extremely helpful!

---

### Decision · Program_Chairs · 2023-01-20

**Decision:**

Accept: poster

**Justification For Why Not Higher Score:**

The technical contribution of the paper is somewhat low for ML conferences. The proposed fix, while intuitive, is somewhat heuristic itself and doesn't come with any crisp theoretical or technical insights / arguments in support.

**Justification For Why Not Lower Score:**

Empirical validation seems comprehensive across challenging problems, reaching SOTA. The authors explore how proposed alignment methods (e.g., whether rank, margin, or reward based) or the choice of the regularization term impact evaluation scores. For example, after alignment, beam size consistently improves performance. The proposed "calibration" recipe also seems superior to fine-tuning in terms of compute times. The value of the paper rests primarily in its empirical results.

**Metareview: Summary, Strengths And Weaknesses:**


The authors propose to use latent decoder embeddings of proposed candidates in comparison to target embeddings so as to fine-tune model likelihoods. This "calibration" step operates on the same fine-tuning data and comes after the regular pre-training and fine-tuning steps. The motivation is to improve decoding qualify as well as to dispense with the need for (many) standard decoding heuristics such as length normalization or beam size selection. The proposed calibration loss consists of alignment scores that involve either rank, margin or reward based scores, each seeking to enforce better correspondence between similarity and likelihood. The calibration loss is accompanied with regularization terms, either cross-entropy or a sample-KL,  so as to keep the model anchored on the basic fine-tuned model. The technical contribution of the paper is somewhat low for ML conferences. The proposed fix, while intuitive, is somewhat heuristic itself and doesn't come with any crisp theoretical or technical insights / arguments in support. That said, empirical validation seems comprehensive across challenging problems, reaching SOTA. The authors explore how proposed alignment methods (e.g., whether rank, margin, or reward based) or the choice of the regularization term impact evaluation scores. For example, after alignment, beam size consistently improves performance. The proposed "calibration" recipe also seems superior to fine-tuning in terms of compute times. The value of the paper rests primarily in its empirical results.


**Note From Pc:**

if the above contains the word "oral" or "spotlight" please see: "oral" presentation means -> notable-top-5% and "spotlight" means -> notable-top-25%. As stated in our emails, we are disassociating presentation type from AC recommendations

**Summary Of Ac-Reviewer Meeting:**